# [Black] Teachers Resisting Damaged-Centered Research: Community Listening Exchanges as a Reciprocal Research Tool in a Gentrifying City

**Thais Council [1],\*, Shaeroya Earls [2],\* , Shakale George [2] and Rebecca Graham [2]**

[1] College of Education, University of Kentucky, Lexington, KY 40506, USA
[2] College of Education and Human Development, Georgia State University, Atlanta, GA 30303, USA; mrs.shakalegeorge@gmail.com (S.G.); grh785@gmail.com (R.G.)
\* Correspondence: thais.council@uky.edu (T.C.); searls2@student.gsu.edu (S.E.)

**Abstract:** Gentrification impacts many cities across the nation. Affordable housing task forces and legislation meant to address housing inequities are becoming more common, yet the authentic experiences of those affected are often unacknowledged. Absent from the discussion of gentrification are the voices of those deeply impacted, some who are at the center of the work to maintain communities: Black teachers, Black students, and Black families. In many school districts, teachers do not have the opportunity to address the systemic issues that impact their students and communities. Still, it is impossible to ignore the ways societal injustice seeps into the classroom. This article discusses our work as a teacher participatory action research collective exploring the intersection of housing and educational displacement in a rapidly gentrifying community in Southwest Atlanta, Georgia. We highlight our roles as community-centered educators and detail how we intentionally and thoughtfully worked to create a reciprocal space to engage communities in Community Listening Exchanges. We present Community Listening Exchanges as a justice-centered innovation to community-engaged research and scholarship. Our critical and collaborative approach to generating and analyzing data allowed us to uncover how housing and educational displacement relies on deficit narratives to justify the removal of marginalized people. We offer CLEs as a reciprocal research tool that deviates from traditional qualitative research and resists anti-Black, damage-centered narratives.

**Keywords:** community listening exchanges; resistance; disrupting damage-centered narratives

## 1. Introduction

We are four community-centered teachers—three Black women and one Jewish-American woman—who teach, live, and work in rapidly gentrifying, Black neighborhoods and schools. This paper details Community Listening Exchanges (CLEs), in which we examined the intersection of urban education reform and gentrification[1] in Southwest Atlanta. Our Southwest Atlanta neighborhood and schools are riddled with poachers—those who lurk from the outside, wait for the prime opportunity (read: return on investment potential), and then pounce on minoritized and marginalized residents who have begged city officials for resources for decades. This is how some in our neighborhood and schools experience gentrification. As community-centered teachers, we too witness and experience gentrification this way. Through our collaborative research, we decided to stand in the gap of urban education reform and gentrification even as it impacts our personal and professional lives. It is with this lived experience that we approach research in our communities. As teachers, we embody community-informed teaching practices, which means we ground our classrooms and ways of being in the expressed needs of students, parents, and communities we serve. As teacher-researchers, we formed a teacher participatory action research (TPAR) collective aptly titled "Teachers at the Center" because we operate within a "double aim" (Fals Borda 2001; McTaggart 1997; Stapleton 2019) in that we seek

to enact social transformation for our students while conducting research through the lens of our communities without a hegemonic gaze. As a method for explicitly centering justice and challenging racial inequity in scholarship, Gordon da Cruz (2017) posits that critical community-engaged research rest on three of six prominent components: examining real-life social problems, engaging in an authentic democratic process to identify issues with community members, and producing knowledge in collaboration with communities directly impacted by the issues. Our paper aims to detail Community Listening Exchanges (CLEs) as a justice-centered, innovative qualitative research method to authentically engage communities and eliminate the hegemonic gaze in traditional qualitative research.

In this article, we assert that our identities as four [Black][2] teachers marginalized by association (Stapleton 2019) with our students and communities, our quest to examine real-life social issues impacting our students and us, and our authentic democratic inquiry process to produce knowledge in collaboration with the communities we serve make our research and activism critical. First, we acclimate the reader with a brief context of Southwest Atlanta and focus on the community in which we live and work. Then, we share a brief legacy of Black teachers who have demanded justice through study and resistance (Kelley 2002) in Atlanta. Next, we chart our democratic inquiry process to define Community Listening Exchanges and share the results of our research. We discuss how insight generated during this process informed educational practices in schools and communities in Southwest Atlanta. Finally, we conclude our paper by highlighting the ways in which Community Listening Exchanges foster trust, compassion, and deeper knowledge in a shift towards justice.

## 2. [Black] Teachers Resisting Damaged-Centered Research

### 2.1. Atlanta as Our Site for Research

During our early stages of research collaboration, we discovered that the voices of those deeply impacted were absent from the gentrification narrative in Southwest Atlanta while also at the center of the work to maintain communities. Those voices are Black teachers, Black students, and Black families. Atlanta, especially Southwest Atlanta, has a rich legacy of Black excellence and mobility. Even with such a rich legacy, Atlanta has been ranked as the city with the highest rate of income inequality in the nation based on a Bloomberg analysis. According to Foster and Lu (2018), many residents struggle with poverty while working in low-paying retail, hospitality, and service industries. As the headquarters for various Fortune 500 companies, the city boasts a 24% poverty rate. Homeownership is another area of inequality in Atlanta, with the lowest percentage of owner-occupied homes belonging to Black residents (Keenan 2019). In Southwest Atlanta, these disparities are often exacerbated. The exclusion of our voices, expertise, and experiences is an injustice, and our teacher inquiry represents a move towards telling the story from the ground. Armed with academic literature, publicly available data, personal experience as residents of Southwest Atlanta, and professional teacher observations in Southwest Atlanta schools, our research collective embarked on an inquiry journey to understand the motivations and desires of Black people and new residents more intimately through CLEs. CLEs offer a humanizing and reciprocal exchange of information gathering and sharing among community members who share similar experiences. In this case, our commonalities lie within the intersection of gentrification and urban education reform in Southwest Atlanta.

Atlanta has long been the test site for public and public–private housing projects. Vale (2013) calls Atlanta a "twice-cleared community" because of the persistent and historical displacement of Black communities twice on the same land. The first public housing project in the United States was Techwood Housing projects initially constructed in 1934 for White working-class residents (Holliman 2008) and later, in the 1990s, again cleared of Black and poor residents for the HOPE VI Centennial Place project in preparation for the 1996 Centennial Olympics (Vale 2013). Currently, Southwest Atlanta is experiencing yet another race and housing clearing that disrupts educational and housing opportunities for Black students and families. Roy (2019) argues that these clearing projects equate to

racial banishment because of the civil, social, and literal death of longtime residents who are banished from their schools, homes, neighborhoods, and communities.

Next, we offer our subjectivities with the research site, not as an ethnographic description, but to highlight how close and "on the ground" we are to educational and housing injustice. Not only are we teachers, administrators, and researchers in Southwest Atlanta, but we each resided in and called Southwest Atlanta home. Rebecca is an Assistant Principal at a turnaround public charter school in a quickly gentrifying neighborhood in Southwest Atlanta. She is the daughter of a 30-year education veteran who, during White flight, opted to stay in her metropolitan Atlanta neighborhood. Shakale is an Atlanta native and the daughter of a veteran educator. Although she works in Southwest Atlanta, her first-time home purchase was well outside the city limits of Atlanta because of limited housing inventory and lack of affordability. Thais experienced pushout when she complained for months of unsanitary living conditions that the landlord was responsible for correcting. Afterward, the landlord refused to offer a lease renewal on a Southwest Atlanta home and, five months later, terminated the month-to-month agreement amid her graduate studies at nearby Georgia State University. Shaeroya "Shae," a Southwest Atlanta native and a teacher at a public charter school in the same area, grew up in Mozley Park—a historically Black neighborhood close to the Atlanta University Center. She is also a Booker T. Washington High School graduate, a Southwest Atlanta neighborhood school with community pride and tradition. Our individual stories made our collective research and action in Southwest Atlanta personal and necessary. At the time of our study, Rebecca, Shae, and Shakale each worked in a turnaround public charter school designed to boost student achievement through a "quick and dramatic transformation" (Cucchiara et al. 2015, p. 261).

### 2.2. Legacy of Black Teachers Resisting Damaged-Centered Narratives

To further contextualize CLEs as an innovation of our TPAR, allow us to share a brief legacy of Black teachers resisting damage-centered narratives in Atlanta. In 1899, African-American educator, historian, sociologist, and faculty at Atlanta University (later Clark Atlanta University), W.E.B. DuBois, learned that a Black man, Sam Hose, had been lynched in nearby Newnan, Georgia. Hose was accused of killing his landlord and raping the landlord's wife. DuBois decided to protest through a written statement intended for the *Atlanta Constitution (now Atlanta Journal-Constitution)*. After learning of the brutality of Hose's murder and the public display of his body parts, DuBois declared that "one could not be a calm, cool, and detached scientist while Negroes were lynched, murdered, and starved." We are inspired by DuBois, whose writing was an act of resistance that disrupted the damaged-centered narrative of his time. This narrative positioned Black people as too afraid or uneducated to confront injustice and ultimately justified their removal (read: murder). He and other Black scholars intentionally used their work as weapons of resistance. Byrd (2016) writes, "Black scholars have always produced significant scholarship and advanced their academic disciplines even as they empowered their communities." Byrd's summation is an extension of Marable's (2000) assertion that Black intellectuals' scholarship offers a prescriptive, descriptive, and corrective blueprint to engage "a practical connection between scholarship and struggle, between social analysis and social transformation . . . for the purpose of transforming their actual conditions and the totality of the society all around them" (p. 18). Much like DuBois, we, too, could not stand idly by witnessing the displacement, erasure, and removal of our neighbors and students from their neighborhood and schools. We are committed to using our positions as teacher-researchers and have decided we must act to transform the experiences of those we share heritage and community with.

Our personal and professional ties to Southwest Atlanta, our shared desire to serve our communities beyond the classroom and curriculum, and our collective love for our students and their families are what propelled us into teacher research and activism via TPAR and CLEs.

### 2.3. A Democratic Inquiry Process to Disrupt Damage-Centered Research

Guishard (2009) describes democratic inquiry as research "collaboratively designed, conducted, analyzed, and disseminated in the context of equal partnerships with university scientists and members of disempowered groups" (p. 87). As a disempowered group—[Black] teachers—the arrival of our collaborative study was haphazard but diplomatic. Our inquiry process started by using our firsthand school and classroom experiences to list the issues plaguing Black students and schools. We considered many of our discussions, readings, and field trips and reviewed transcripts of our previous work sessions. An initial transcript review from our TPAR process revealed the number of times we referred to housing, renting, homelessness, or gentrification. After several back-and-forth discussions laced with dissent and difference, we settled on gentrification as the topic of our collaborative research study:

Shae: We could do another group like this but with parents. Because we gotta disrupt the knowledge. A lot of parents have the same Eurocentric beliefs about school and attendance policies because that's what they've been taught. But if we disrupt their knowledge about it, then they could probably start burning a fire. That would be the first thing though. We would not be able to do anything else successfully until they're informed.

Shakale: Right. I like the article that I read. I liked the community part and the vocational part. I think this is kind of going back to us seeing how we can market what our skills and the things that we do in our communities. How to show people how to make money using their gifts and their talents? I think the main problem is that poverty piece. So, my personal thing I think we should focus on is that poverty piece.

Rebecca: I don't know. I mean, I think the poverty thing, I think that's too. I don't know. I feel like that's, like, "You all in poverty." You bringing people in, "You know you all in poverty, right?" [laughter] That's how I feel like. I think what Shae said, inviting parents and focusing on [something] not too much, like in your face. And then get people talking about like what you said about what parents need. Asking them questions but not to get them to say like, "Oh, we want our children to go to college," or something like that but something more with the school, just have a conversation.

Thais: So, [Rebecca] asked a question, "What's the end goal?" So, in my brain, the end goal is what question am I trying to ask and answer? What am I trying to find out by doing this? Yeah, we know we've gotta disrupt knowledge but we're trying to disrupt knowledge to get to what point? What's our aim, and that's something we may have to like continue to think about. Because there are layered issues. But what's the one pressing issue for us collectively? So, what I hear us talking about a lot and just reading the transcripts . . . We talked about housing a lot. How much housing is impacting the, the "what" of what's happening in our schools, how we're doing our jobs and even why we show up sometimes.

From this discussion, one could deduce that we desperately wanted to use our social analysis as a tool for social transformation. Discovering our students', families', and communities' struggles was personal and political for each of us. We also could not deny that our neighborhood was changing all around us. There were new school buildings, renovated houses, new houses, new parks, new businesses, more police presence, foregrounded that Black students were withdrawing from school and moving each week. Several discussions included statements such as:

Shae: So, I went to Washington [High School] and it's extremely gentrified now but the enrollment of our school now is so low because all the White people who live over there, their kids don't go there and they're not planning on sending them there.

Shakale: Yeah, that bothers me to no end. Now that I'm there, those are my children that I love. I don't want to lose them, and they're starting to move already. I've lost three students in the past few months because they got "that letter" and at the apartment complex, rent is increasing. That's hearsay. I never saw the letter but that's what the word on the street is and that's why a lot of students are moving.

Thais: So, I got a postcard in the mail from a neighborhood resident asking us to call the police on drug dealers. The so-called drug dealers are children.

All of the markers of gentrification surrounded us: changing demographics of our schools, letters from landlords and investors, and postcards from newcomers who saw Black children as a threat. We read articles, watched documentaries, and discussed the school-to-prison pipeline (STPP) and realized that the STPP was nuanced with gentrification, adding another provocation to school pushout. We theorized that the school system, real estate holders, and developers were collaborating to create a neighborhood free of poor, Black people and that gentrification is the new school pushout that led us to examine the experiences of our longtime and new neighbors more intimately.

Our democratic inquiry was not unlike a typical inquiry project in that we followed a process to conduct our collaborative study. We started by reviewing the literature on the intersection of urban education reform and urban education reform. We found that the current literature (Davis and Oakley 2013; DeSena 2006; Housing Justice League and Research Action Cooperative 2017; Hankins 2007; Lipman 2011) did not offer a glimpse into the intersections of urban education reform and gentrification through a teacher inquiry.

To begin our data collection process, we adhered to the requirements of the Institutional Review Board, and each co-researcher completed the human subjects Collaborative Institutional Training Initiative (CITI) certification to be recognized as formal researchers. We each contributed to and co-authored a formal research proposal. We detailed our research design, research questions, data collection methods, data analysis, target participants, recruitment methods, length of study, and a plan for presenting findings. What made our inquiry process democratic was our way of thinking and approaching the study and the dissent and consensus displayed throughout the process.

When initially designing our collaborative study, we aimed to intimately understand the factors and decisions impacting the housing security of our Black students and families. We talked, debated, and collectively decided to conduct one-on-one semi-structured interviews. We discussed our relationships with parents and how we could harness those relationships to ask parents hard questions. We even agreed that one-on-one interviews are more intimate than surveys. We developed an interview protocol with primary and deeper probing questions (see Figure 1).

Our initial design was a mistake. Our semi-structured interview approach amounted to unintentionally presenting ourselves as social workers instead of thoughtful listeners and intentional researchers. Our goal was to interview parents to survey their living arrangements in a rapidly gentrifying neighborhood. Above all, we wanted to know if parents felt safe and comfortable or dismissed and disposable in a shifting neighborhood. We created a list of potential interviewees and anticipated responses, hoping to gain insight into the effects of urban education reform and gentrification. Ultimately, we did not conduct interviews and consulted our elders instead.

We often consulted elders and our community partner, Gwinnett SToPP, for guidance and feedback. A conversation with Dr. Joyce King, Thais' doctoral advisor, revealed the voyeuristic nature of the initial interview protocol. Dr. King explained that community-centered work must place the community's lives, experiences, and perspectives at the center of the work and rethink our approach through the participants' eyes (King and Mitchell 1995). She challenged us to deeply reflect on our end goal and the process to reach the goal. She also challenged us to think about the damage and trauma many of our parents have experienced and how our research could either add to the trauma or disrupt the damage.

Essentially, the interview protocol perpetuated anti-Blackness where we would have triggered our parents and created distance in our relationships. We questioned how we would feel if a group of well-intentioned teachers asked to interview us and asked questions about our education, earnings, and knowledge of homebuying? This process forced us to be deeply reflective and cognizant of any damage-centered narrative we would have projected. Our reflexivity was constant throughout as we often brought our professional lens as teachers to our studies while also remaining mindful that our students and families may have had harmful experiences with schools and teachers. In this way, the double

aim of participatory research helped us reflect on ideological thinking and prepared us to collaborate with communities in humanizing ways.

---

*Interview Protocol A (Parents/Guardians)*

1. Do you own/rent/live with someone?
    a. If you rent, would you like to become a homeowner?
        i. If so, do you know the steps to becoming a homeowner?
    b. How much do you pay per month for housing?
        i. Is this amount affordable or out of reach for you?
    c. If you are not a homeowner or renter, who do you live with?
    d. How long have you lived here?
2. Do you consider yourself low, middle, or upper class?
    a. Do you mind sharing how much money you make per month?
3. How do you feel about your child's school?
    a. Why do you send your child to this school?
    b. If you could choose where to send your child to school, where would you send them? Why?
    c. Do you think the school superintendent, district, and principal hear your concerns about the school?
    d. Do you think school leaders and decision-makers are accessible?
    e. Do you know how to voice your concerns and get action?
4. What is your primary mode of transportation?
    a. How reliable is your transportation?
    b. How accessible is your job?
    c. How accessible is your school?
    d. If you lost your transportation, how would this impact your job/school?
5. Describe the climate of the school you/your students attend?
    a. Is it welcoming?
    b. Are you greeted properly?
    c. Do school staff members and teachers ask you for your opinion and input?
    d. Who brings you?

---

*Interview Protocol B (Students)*

1. Who do you live with?
    a. How long have you lived with this person/these people?
    b. When was the last time you moved?
    c. How many people live in your house?
    d. Do you have your own room?
    e. Who do you share a room with?
2. How do you like your school?
    a. Are you ever late or absent from school?
3. How do you get to school?

---

**Figure 1.** Interview protocols.

As a result of our collaboration with our elders and our reverence for the community, CLEs evolved organically. We proposed a Community Listening Exchange instead of interviews or surveys because we saw it as a reciprocal data collection method where participants would provide us with data. In turn, we would share information and resources with them. Our democratic inquiry process demonstrates our strong connection to the community, which was evident in how we sought to privilege the knowledge and experiences of those we serve instead of relying on dominant narratives. We acknowledged and understood that our literature review was a critical process where we interrogated and critiqued each text we read through the lens of our students, families, and communities. We also considered our lived professional and personal experiences when critiquing and interrogating the literature on the intersection of urban education reform and gentrification. We privileged our target audience's knowledge and experiences, constantly reminding ourselves that we wanted information they provided us versus what literature says about them. Lastly, because we saw our participants as valuable assets to the community and our research, we offered compensation through gift cards to frequented merchants such as gas stations, grocery stores, and discount stores. Dr. King's critical feedback, coupled with our reflexivity and critical discussions, led us to conduct CLEs designed for the community to gain resources while providing data through collective group thinking, reflection, and dialogue.

### 2.4. Community Listening Exchanges

Tuck (2009) charged researchers to "institute a moratorium on damage-centered research to reformulate the ways research is framed and conducted and to reimagine how findings might be used by, for, and with communities" (p. 409). Additionally, Quarles and Butler (2018) assert that "integrating multivocal literatures into school gentrification research is critical to interrogating a policymaking context" which includes the lived experiences and expertise of those "on the ground".

In the summer of 2018, our TPAR collective conducted three CLEs, with three distinct communities, in living and dining rooms across Southwest Atlanta, Georgia. The goal of each listening exchange was to listen to and exchange knowledge with communities impacted by the intersection of urban education reform and gentrification in Southwest Atlanta. Through CLEs, we sought to counter Tuck's (2009) definition of damage-centered research—research that intends to document peoples' pain and brokenness to hold those in power accountable for their oppression (p. 409). Instead, CLEs allowed us to privilege our communities' knowledge and experiences while charting their brilliance, community commitments, and historical excellence.

King and Mitchell (1995) posit that research with, by, and for Black people should follow an Afrocentric methodology. This methodology is committed to liberation and offers a way to decenter the center (read: hegemonic gaze). Additionally, Afrocentric methodology allows Black people to see the world from the center of their own being, accepting their stories as accurate through a collective Black experience. CLEs gave us the freedom to generate liberating self-reflexive knowledge about our everyday lived experience away from the eyes of researchers examining and critiquing our lives from a distance. CLEs are a "practical-critical activity" (King 2019) whereby "cultural amnesia, dysconscious racism, and (mis)educating pedagogies of oppression in schools, popular culture, corporate media, and other societal processes that alienate us from . . . our spirituality and humanity" (p. 15) are disrupted. CLEs operate through this "practical-critical" praxis hence offering an innovative approach to community-engaged research while magnifying the multiple assets and collective power of marginalized communities. CLEs differ significantly from traditional focus groups because of the intentional approach of the researchers. For example, our teacher PAR collective went into CLEs to share and interact as "participants." After posing questions, we often shared our own experiences as a sort of "altar call" (Powell and Coles 2020), inviting others with similar experiences or opposing views to air their experience. We did not believe we had all the answers, and we were not there to quiz

anyone on their knowledge. Focus groups can sometimes mirror a fishbowl where the researcher sits outside and watches how the conversations go and sticks their finger in to stir it up every now and then. In a Community Listening Exchange, researchers, community members, and attendees are in the bowl together, examining our collective experiences. In essence, CLEs allowed us to move beyond current literature (Davis and Oakley 2013; DeSena 2006; Housing Justice League and Research Action Cooperative 2017; Hankins 2007; Lipman 2011) that focused solely on gentrifiers entering neighborhoods and schools into an approach that centered the experiences of Black students and families who were being displaced, unable to benefit from the changes in their schools and communities (Cordova-Cobo 2019; Quarles and Butler 2018).

This paragraph offers a general overview of each Community Listening Exchange to prime the reader for further details in the next section. Each Community Listening Exchange aimed to examine how gentrification and urban education reform intersect in Southwest Atlanta. When deciding how to recruit for the first listening exchange, we were careful to consider the digital divide in the neighborhood, meaning some new residents communicate solely through digital devices and social media platforms. In contrast, longstanding residents typically talk face-to-face or by phone. Because we desired to have a balanced representation of participants at the listening exchange, we decided to rely on our community partners and word-of-mouth tactics to recruit participants. Additionally, each co-researcher distributed flyers (see Figure 2) at our respective schools and neighbors. The first exchange took place in the open concept living and dining room of a renovated house in Pittsburgh, Southwest Atlanta, a historic, working-class, Black neighborhood. Participants included homeowners, renters, teachers, students, real estate professionals, and investors looking to purchase in a rapidly gentrifying Atlanta neighborhood. During our data analysis of the first listening exchange, we discovered that the voices of White gentrifiers were missing from our investigation. Because we are careful researchers, we did not want to misrepresent their views. Ultimately, we decided that a second listening exchange solely focused on White newcomers was necessary. We hosted the second exchange in the living room of a Southwest Atlanta neighborhood family new to the neighborhood. White and biracial nuclear families with school-aged children comprised the entire audience. We discovered that we had not reached data saturation during our data analysis of the second listening exchange, particularly because Black parents' voices were absent. The third and final exchange happened through a longstanding nonprofit organization and was convened in the dining room of a house converted into an after-school space and office. The participants were all Black women with children or grandchildren who attended the local public charter school. There were 37 Black and White participants across all exchanges whose ages ranged between twelve to eighty years old (see Tables 1–3). In our roles as teacher-researchers, we were not immune to the threat of being negatively impacted by Georgia's employment-at-will policies[3]. To be present as teacher-researchers while understanding threats to livelihood, all sessions were not held in school buildings but at a neutral location to allow teachers and participants to speak freely. The informed consent form included information on the dangers of exposing our collective conversation via social media. All community members and attendees were asked not to take pictures during the exchange or post the exchange content online.

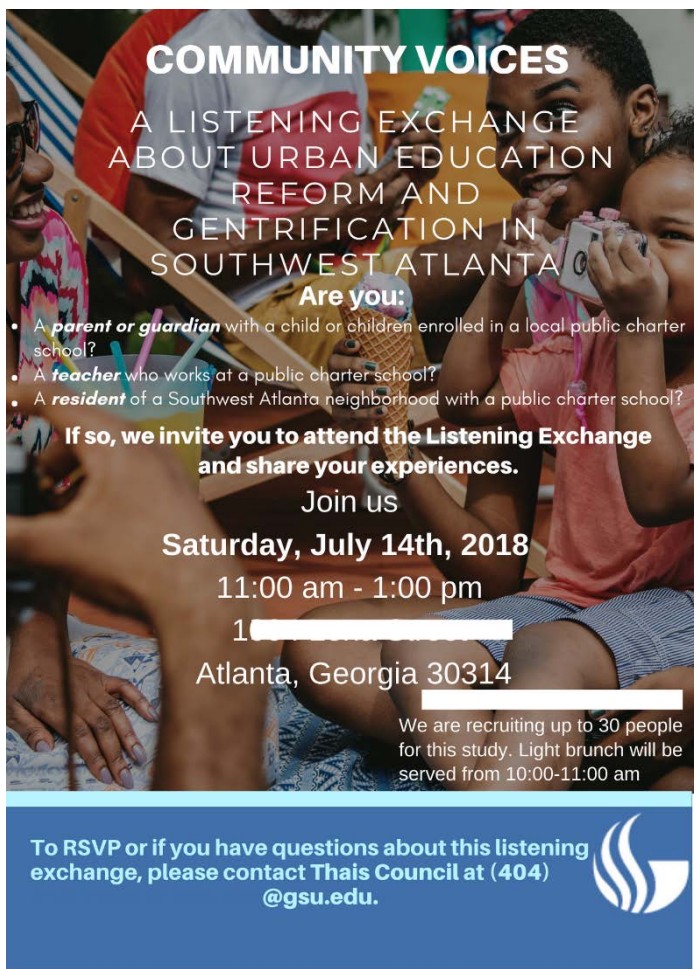

**Figure 2.** CLE recruitment flyer.

**Table 1.** Listening Exchange 1 participant profiles.

| Participant | Gender | Age Range | Race | Neighborhood |
|---|---|---|---|---|
| #1 | Female | 35–44 years old | Black | Capital View |
| #2 | Female | 35–44 years old | Black | N/A |
| #3 | Male | 45–54 years old | Black | Mableton |
| #4 | Male | 25–34 years old | Black | N/A |
| #5 | Male | 25–34 years old | Black | N/A |
| #5 | Male | 25–34 years old | Black | Norcross |
| #6 | Male | 45–54 years old | Black | N/A |
| #7 | Female | 35–44 years old | Black | South Atlanta |
| #8 | Female | 45–54 years old | Black | Norcross |
| #9 | Female | 35–44 years old | Black | N/A |
| #10 | Female | 25–34 years old | Black | Hammond Park |
| #11 | Female | 25–34 years old | White | Hapeville |
| #12 | Female | 25–34 years old | Black | West End |
| #13 | Female | 35–44 years old | Black | Hammond Park |
| #14 | Male | 45–54 years old | Black | Adair Park |
| #15 | Female | 45–54 years old | Black | N/A |
| #16 | Male | 12–17 years old | Black | N/A |

**Table 2.** Listening Exchange 2 participant profiles.

| Participant | Gender | Age Range | Race | Neighborhood |
|---|---|---|---|---|
| #1 | Female | 35–44 years old | White | West End |
| #2 | Male | 35–44 years old | White | West End |
| #3 | Female | 25–34 years old | White | Ashview Heights |
| #4 | Male | 25–34 years old | Black | Ashview Heights |
| #5 | Female | 25–34 years old | White | Washington Park |
| #6 | Female | 25–34 years old | White | Washington Park |
| #7 | Male | 25–34 years old | White | Washington Park |
| #8 | Male | 35–44 years old | White | Adair Park |
| #9 | Female | 25–34 years old | White | Hapeville |
| #10 | Female | 25–34 years old | Black | Hammond Park |
| #11 | Female | 35–44 years old | Black | South Atlanta |
| #12 | Female | 35–44 years old | Black | Capital View |

**Table 3.** Listening Exchange 3 participant profiles.

| Participant | Gender | Age Range | Race | Neighborhood |
|---|---|---|---|---|
| #1 | Female | 35–44 years old | Black | Capital View |
| #2 | Female | 25–34 years old | White | Pittsburgh |
| #3 | Female | 35–44 years old | Black | Pittsburgh |
| #4 | Female | 25–34 years old | Black | Pittsburgh |
| #5 | Female | 25–34 years old | Black | Pittsburgh |
| #6 | Female | 35–44 years old | Black | Pittsburgh |
| #7 | Female | 55–64 years old | Black | Pittsburgh |
| #8 | Female | 25–34 years old | Black | Pittsburgh |
| #9 | Female | 25–34 years old | Black | Pittsburgh |

**Community Listening Exchange 1.** The first listening exchange was coordinated with our community partner, Gwinnett SToPP, and a local realtor and executive director of a housing nonprofit, Ronald Denson (pseudonym). We initially formed a relationship with Denson to gain some insight into Southwest Atlanta's housing patterns from a real estate investor's perspective. Although he was skeptical of some of the social and political implications of the housing crisis, he felt motivated to support our work. He helped us secure a vacant, newly renovated house owned by the Annie E. Casey Foundation in Pittsburgh, Southwest Atlanta. Denson also donated food and drinks for the listening exchange. Because this was a vacant house, Denson forgot to turn on the air conditioning. It was an early June afternoon in a city affectionately known as "Hotlanta," and the living room did not shed Atlanta's namesake. Even with several fans blowing, the living room was a literal sweatbox. Every participant, however, stayed until the very end. A white projector screen was positioned at the far end of the living room to display our presentation and listening exchange questions (see Figure 3). To maintain the reciprocal commitment of the listening session, the co-executive director of Gwinnett SToPP introduced the school-to-prison pipeline phenomenon and how Georgia education policies deepen disparate outcomes for Black children and Black communities. We invited local Atlanta historian and owner-operator of Black Mecca of the South Tours, Nasir Muhammad, to provide a brief history of Southwest Atlanta and the Pittsburgh community (see Figure 4). The in-kind food donation, defining STPP, and the Black Mecca of the South Tours history lesson was our way of giving more than we took from participants. These contributions (both in resources and information) are crucial in CLEs because, as researchers, it is our responsibility to be aware of the ways marginalized communities are used for information and left with nothing in return.

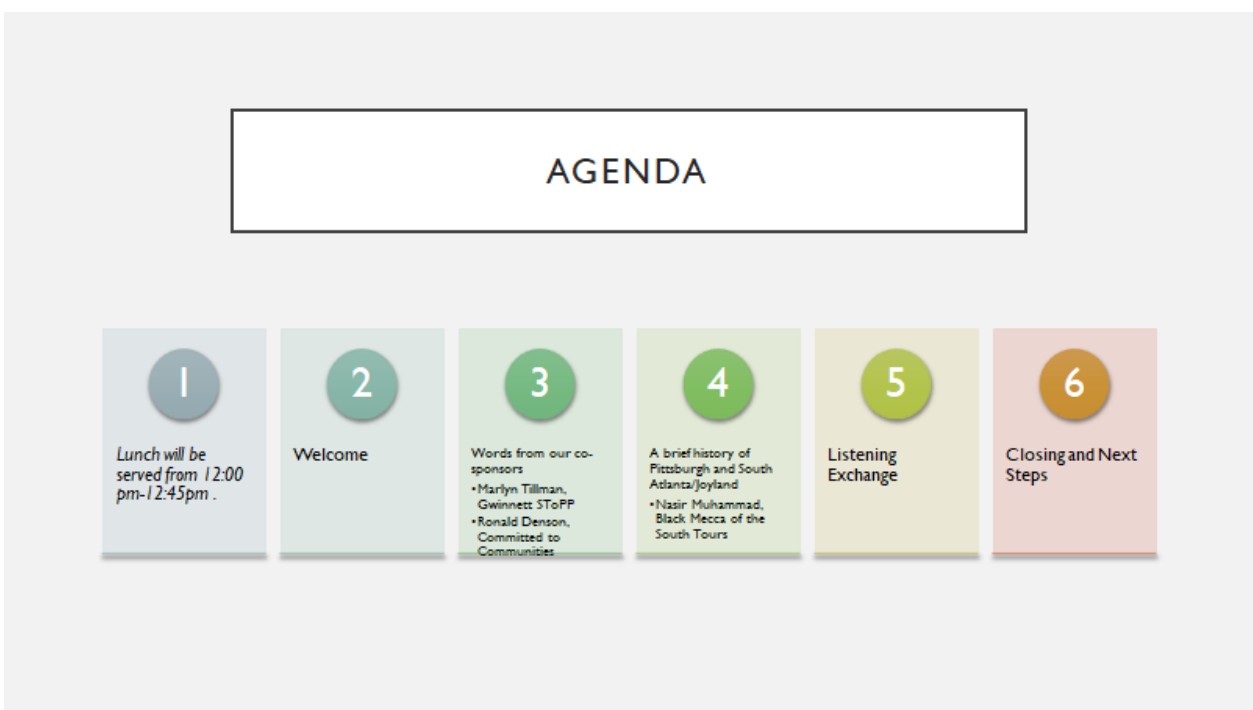

**Figure 3.** Community Listening Exchange PowerPoint agenda.

Another consequence of CLEs is sharing knowledge and correcting misconceptions that naturally occur amongst attendees. In the first listening exchange, participants ranged in their backgrounds and experiences, and they leaned on those contexts to contribute to the discussion. For example, one participant, a Black man from a middle-class background, discussed his shock when his daughter's affluent, well-performing public school was filled with Black teachers: "*One of the things that I found interesting about [affluent white school] is a lot of the teachers were Black and a lot of the students were not Black. But they had great education and so it looked like a [sic] oxymoron to me because I expected, I expected that all (pause) most of the teachers were gonna be White and that wasn't the case.*" His expectation that Black students and Black teachers would be incapable of a "great education" was steeped in anti-Blackness. Still, without prompting from moderators or a lesson on ingrained White supremacy, his views were countered by another participant's comments. A Black woman, raised and educated in Southwest Atlanta, recounted the high expectations and quality education she received from Black educators: "*I grew up in my ideal neighborhood. I grew up where my parents were one of the first Black families to move into Cascade. My parents took over West Manor if you look at the pictures from 1964, West Manor class of 1964 it was an all-white school. By 1972, it was [a] predominantly Black school with Black teachers. APS [Atlanta Public Schools] left us alone. APS never came over to West Manor Elementary School: "Leave those Black kids alone. Because those parents do not play, leave them alone, whatever they want." We had music, art, Spanish, PE every day, music every day, art every day. All the stuff they say happens on the north side, we had on the south side at West Manor. They left us alone. We tested off the grid. And we did well. And we had a community around that school that supported that school.*" As illustrated, CLEs provide space for insightful discussions where all participants' voices are valued, where no person's knowledge is privileged, and where everyone has the opportunity to give and receive.

**Community Listening Exchange 2.** The second listening exchange was in the living room of a White family new to Washington Park, Southwest Atlanta. This family had been in the neighborhood since 2009. Lee (pseudonym) worked as a missionary and basketball coach at nearby Booker T. Washington High School—an all-Black, historic neighborhood school where Dr. Martin Luther King Jr. matriculated—and knew one of the co-researchers since her teenage years at the school. Lee also works with a local foundation. Lee recruited her circle of friends and neighbors using her strong neighborhood ties. Our TPAR collective

supplied bagels, cream cheese, fruit, and coffee for this listening exchange. Instead of setting up a white projector screen, we used blank post-it chart paper as our backdrop. We positioned several urban education and urban housing books across the coffee table. We provided each participant with a folder with the consent form and several articles related to gentrification and urban city schools (Buntin 2015; Desmond 2018; Dill et al. 2016; Fenwick 2013; Hannah-Jones 2015; Kirk 2018; Quinlan 2016; Underwood 2018). One of the participants was the dean of a charter network. He asked to take a picture with us, and Thais declined. His response was, "That's smart to stay out of pictures when you're doing controversial work." We noticed a vastly different conception of urban education, gentrification, and housing in Atlanta during this listening exchange. Most participants were not native to the city, many moving into the neighborhood within the last ten years. All but one participant were White.

---

*Pittsburgh* was established in 1883. Pittsburgh is directly connected to Atlanta through the railroad industry. Pittsburgh is one of the oldest African American communities in Atlanta. African Americans in Pittsburgh gained employment through three railroad lines that cross the neighborhood: East Tennessee, Virginia, and Georgia Railroad Company. Pittsburgh, along with an area called Joyland and South Atlanta is where Black money lived. This is also the area where Clark College was established and originally located so Blacks who lived in these areas were connected to the institution. That is why you can go down close to the railroads and travel down through South Atlanta and most of the streets are named after people who were connected to old Clark College.

*Carrie Badger Pittman*. She had the distinction of being one of the few women who graduated from Atlanta University and also Morehouse. Carrie Pittman's husband, John Pittman, was a Morehouse graduate in 1926. Carrie Pittman graduated from Morehouse at the age of 61. Pittman Park in Pittsburgh is named for her. But one of the things we noticed in Atlanta, when something is named after a woman, for the most part, you never see a picture, you never see a monument, never see a statue. We have like five or six statues of Dr. Martin Luther King, Jr. I am not saying we should not have them. I am suggesting that there should be some balance. Carrie Pittman was the principal at Roach Street Grammar School which is where Blacks were educated and also William H. Crogman School which is of course, now is Crogman Lofts.

*Carrie Steele Logan*, born into slavery and orphaned as a child, used her $100 a month salary as a stewardess for the Central Railroad to buy land and build the Carrie Steele Orphanage. The orphanage started on Auburn Avenue but it was so small in that little wooden house. She ends up purchasing this big property right here in Pittsburgh which is the same location of where Gideons Elementary School is right now. That was her whole block right there and her school ends up leaving this area and going to the west side where it is now. It is the oldest continuously operating Black orphanage in the United States of America. None outlasts this one and it is from a Black woman who was in this area of Pittsburgh, working hard to actually do that. No marker for her except the one on Auburn Avenue.

The original alma mater song of **Clark College** starts off with, "There's a school on a hill." Now you know, right now, Clark Atlanta University, it don't sit on no hill, but if you go over here to Carver High School, that's a hill. The original Clark College was one of the greatest architectural buildings in Atlanta. Black men built Clark College from the ground up. Where Carver High School sits now is the old Clark College, now known as Clark Atlanta University. Clark College had twelve to fifteen properties on campus. Some of the buildings were named after local

---

**Figure 4.** *Cont.*

*Black luminaries (Warren Hall, Christman Hall, and Crogman Chapel), while other buildings were named after Caucasians from the north who supported the institution.*

*What we have to begin to understand is that the Pittsburgh community, the south Atlanta community, and the Joyland community, were all connected. The brain power, the education, the money, in Black Atlanta all came out of this area.*

***Charles L. Gideons*** *is also a part of Pittsburgh history. He was a longtime school administrator with Atlanta Public Schools and Gideons Elementary School is his namesake.*

***George Washington Carver*** *doesn't have a school named after him by accident. George Washington Carver used to come to this community right here and if you know where the property Annie Casey has right now, the new development taking place, that was farmland. The students at Clark had to work the farm to make money to stay in school, otherwise, they'd be sent back home. George Washington Carver came here in the 1890s after Booker T. Washington gave his famous speech at the Cotton Exposition. George Washington Carver began to teach them how to cultivate the land, produce, stay in school, and make money selling their produce. He also comes in 1923 and continues to look at their efforts and ensure they're doing it. How do I know that? Because I have letters from Bishop Henry McNeil Turner talking about George Washington Carver coming to Atlanta.*

*Now, another thing that Pittsburgh does not get credit for is the refuge the community provided after the **Atlanta Massacre in 1906**. Don't let anyone ever tell you that Sherman burning down Atlanta during the Civil War was the worst thing that ever happened in terms of fire. No! That's not the worst fire in Atlanta's history. The Great Atlanta Fire of 1917 from Edgewood Avenue to Ponce de Leon Avenue left 10,000 Black people homeless. The Pittsburgh community opened its doors. Pittsburgh churches, communities, and organizations allowed Black families from the Old Fourth Ward to come and live. Ebenezer and Wheat Street Baptist Church burned down during the great Atlanta Fire, and so much of the history and documentation was lost in the fire. So, you had a lot of Blacks who were left homeless in this situation, and Pittsburgh served as a safe place for them. Now, I offer two tours that explain this history. One is on Sweet Auburn Avenue, and one is right here in Pittsburgh.*

*We say gentrification, but there's another type of radical element that's racist in nature: the **building of interstate highways**. Once they built I-75, I-85, and I-20, it decimated the community. Many people tried to look at the residents, but the residents reflect those in leadership who were passing laws and regulations to decimate their community. The southern end tip of Pittsburgh was completely cut off when they built I-75, I-85, and I-20. Yes, people are able to move around, but you notice that interstates are not built through White neighborhoods. And the way stadiums are built all across the country, all of these things are directly connected to severing Black communities, and so this happens in Atlanta; it happens in Pittsburgh. And unfortunately, only a few people were compensated when they built I-75 and I-85, and that was a few Black women who had property on Auburn Avenue. Most people were displaced by eminent domain or forced completely out* (First Listening Exchange Transcript).

**Figure 4.** Nasir Muhammad, local historian and owner-operator of Black Mecca of the South Tours, "Leg-acy of Southwest Atlanta".

Because of the different racial composition of the participants in the second listening exchange, we posed an additional question that was not a part of the first listening exchange: Do you consider yourself a gentrifier? Why or why not? One of the White male participants stated: "*But when we moved in, I didn't even know the Beltline was down there and that's just been a beautiful, wonderful thing. I know it comes with other issues but I think the main draw for loving this neighborhood is the community, the Beltline. I think everybody is moving or wanting to move towards like a more, like walkable, bikeable lifestyle and a neighborhood like this offers that and I think the part, why I said I don't feel like I'm a gentrifier right now is the profit. We don't have this five-year plan, wait until our house is worth whatever amount and all right, see*

*you later and taking our money and running. We don't have a plan to ever move. So if I think of gentrification as being profit-driven and I disagree with feeling that way for myself but yeah, I don't know. I don't know if I'm just stuck in my middle-class ways, oh, yeah, our street looks nice and, you know, it was bank owned. It was an empty house, too. So we didn't just, and you didn't displace anyone necessarily either."* Participants in the exchange also discussed their ideas of the neighborhood public schools. Most of them had enrolled their children in charter schools or Montessori schools that they created amongst their networks. We wanted to understand what experience they had with public schools, what they knew about the schools in the area, and what was stopping them from utilizing public schools even though they chose to live in the neighborhood. One White female participant stated: "*I live in the neighborhood that's on the other side of that street that's called Ashview Heights. And we have two little girls and they are at a Montessori daycare in the neighborhood in Ashview Heights, called [school name redacted]. And what brought me here today? I'm a former teacher and I work in the school systems. I've also noticed how quickly our neighborhood is changing. I've been here more than ten years, but I don't quite know how the schools are changing. And to be honest, before I had kids and I lived over here, and I was teaching in Vine City. I was like oh, my children won't go to the public schools in the neighborhood. They won't be there and then I had kids and I was like not quite sure. And so, I want to be there and I want to be a part of the community's changing schools in my heart. My head is in a different place*".

The insight and perspectives shared during the second CLE were invaluable. We understood the importance of relationships and trust to engage in what one participant considers "controversial work." We also knew that it would be critical to include the perspectives of gentrifiers in this work. Because of the reciprocal nature of CLEs, we relied on the resources, books, and articles we provided as an opportunity to exchange knowledge with our participants. Additionally, as a mostly Black research collective entering the home of White participants, we wanted to honor the trust they had extended to us and not interrupt when they shared stereotypical views.

**Community Listening Exchange 3.** The third listening exchange was more informal than the previous two. The Stewart Center is a local non-profit in Pittsburgh, Atlanta, and has long served students from Gideons Elementary School. The Stewart Center After-School program coordinator and Thais have a longstanding relationship. In fact, Thais coached teachers who tutored Stewart Center students as part of her teaching internship in the Georgia State University Urban Literacy Clinic under the direction of Dr. Amy Seely Flint. Amber (pseudonym), a White female and the program coordinator, mentioned that the Stewart Center partnered with Annie E. Casey Foundation to renovate and rent houses to Gideons Elementary School parents in an effort to keep them in the neighborhood and neighborhood school. Many of the parents were Black women who could not otherwise afford to purchase a home in the rapidly gentrifying neighborhood, and one Black grandparent who owned a home with her husband in the neighborhood. Amber extended an invitation to a Thursday night meeting where parents and guardians convene weekly to discuss issues impacting their families, including housing. Amber mentioned housing was a pressing issue, and parents would love to talk with university researchers about it. The exchange was hosted in a house converted into an after-school site at a dining room table with Black mothers and grandmothers. This exchange functioned as a conversation and did not include a presentation or predetermined questions nor was it recorded. It was an informal session, where no notes were written while people spoke. This was a deliberate act to place an emphasis on listening intently to their concerns. We encounter these parents often through our school interactions at Gideons Elementary School, and their voices are often dismissed. Because of this, we did not want to use an out-of-school setting to talk "at" them; therefore we were fully engaged to remain present and intentional. After the session, we wrote journal notes to capture recollections of what was discussed. The women were fully aware of the resources available to them in their neighborhood and school. Gideons Elementary School provided them with a list of rental properties available each week. One parent acknowledged that the list "never included houses for

sell." In addition, this listening exchange was held on a weeknight and not a weekend day like the previous two, and all of the participants were working (grand)parents. We were intentional to honor their time; therefore this exchange lasted an hour instead of two hours like the previous two exchanges. During the discussion, (grand)parents shared their actions against gentrification and leaned on us for academic resources for their children. One of the mothers capped the conversation saying, "I'm not going to let anyone kick me out. I'm going to fight for my right to stay".

### 2.5. Resisting Damage-Centered Research beyond Community Listening Exchanges

Earlier, we referenced leaning on the legacies of Black intellectuals as a framework for analyzing and interpreting Black life and a framework for moving our social analysis to social transformation. As Black teachers in Atlanta—especially after the sensational Atlanta Cheating Scandal that publicly criminalized 35 Black educators—we rest on the fact that our research is our action. We were invited to several venues to share our research and findings and contextualize gentrification through an educational lens, showing our audiences how the intersection of urban education reform and gentrification directly impacts students and families. We presented our paper, "Is Gentrification the New School Pushout: A Conversation with Community about the Intersection of Urban Education Reform and Gentrification," at the 11th Annual Sources Conference sponsored by the Alonzo A. Crim Center at Georgia State University. The room was filled with teachers, and there was standing room only. Many attendees shared similar stories of students and families withdrawing from schools and not returning. We also headlined the Black Education Network (ABEN) Regional Conference, "Follow the Drinking Gourd to Atlanta," at the Auburn Avenue Research Library. This presentation was packed with community members, teachers, students, and several people impacted by gentrification. The audience was so invested in what we had to share that we did not make it through a third of the presentation. Additionally, a Clark Atlanta University (CAU) political science professor heard of the ABEN presentation and invited us to present at the Urban Politics and Policy Seminar at CAU. During this presentation, we were able to engage with policy students and other scholars who had not considered the ways education was impacted by housing displacement. Lastly, we were invited to appear on the "On Point with Juandolyn Stokes" WAOK public radio program, where we detailed for listeners the impact of urban education reform and gentrification in Southwest Atlanta. We share this to illustrate how the insight generated in CLEs found its way into and informed educational practices in schools and communities in Atlanta. As teacher-researchers, we extended our educational reach well beyond the classroom and began to inspire other teachers, scholars, and activists to consider the impact of housing, gentrification, and urban education reform on Black students and families in Southwest Atlanta.

### 3. Data Analysis

Our article details our review and assessment of an innovative community-focused data collection method. Data analysis was a key component of our collective research process in that it highlights how CLEs yielded quality and authentic data in large part to the intentionality of CLEs. During our Community Listening Exchanges, we explored participants' understanding of and experiences with gentrification occurring in their schools and neighborhoods. Specifically, we wanted to understand how they would describe the changes happening in their schools and community, how they felt about the changes, and how they saw themselves and their families within the changes. Our data analysis process included four iterative steps (see Figure 5). We started by reading the transcripts from each session and writing analytic memos to capture our initial thoughts. Next, each co-researcher shared their interpretation of the raw data while the others listened and responded to either ask a clarifying question or deepen their own understanding. Once all researchers shared, we collaborated to create emergent codes that reflected our initial findings from the data. After further reading, analyzing, and discussion, we identified

two main categories in which our codes could be sorted: anti-Blackness and resistance. One of our goals during this process was to analyze the data through a Black Intellectual, asset-based understanding. As teachers and teacher-researchers, the data analysis process was part of our individual and collective learning. We also developed a deeper, more critical lens of our teaching, learning, and leadership practices. Not only did this inquiry raise our awareness, consciousness, and voice as teachers through the data collection and analysis process, but many of the themes we identified also intersected with our personal and professional lives as teachers and Southwest Atlanta residents.

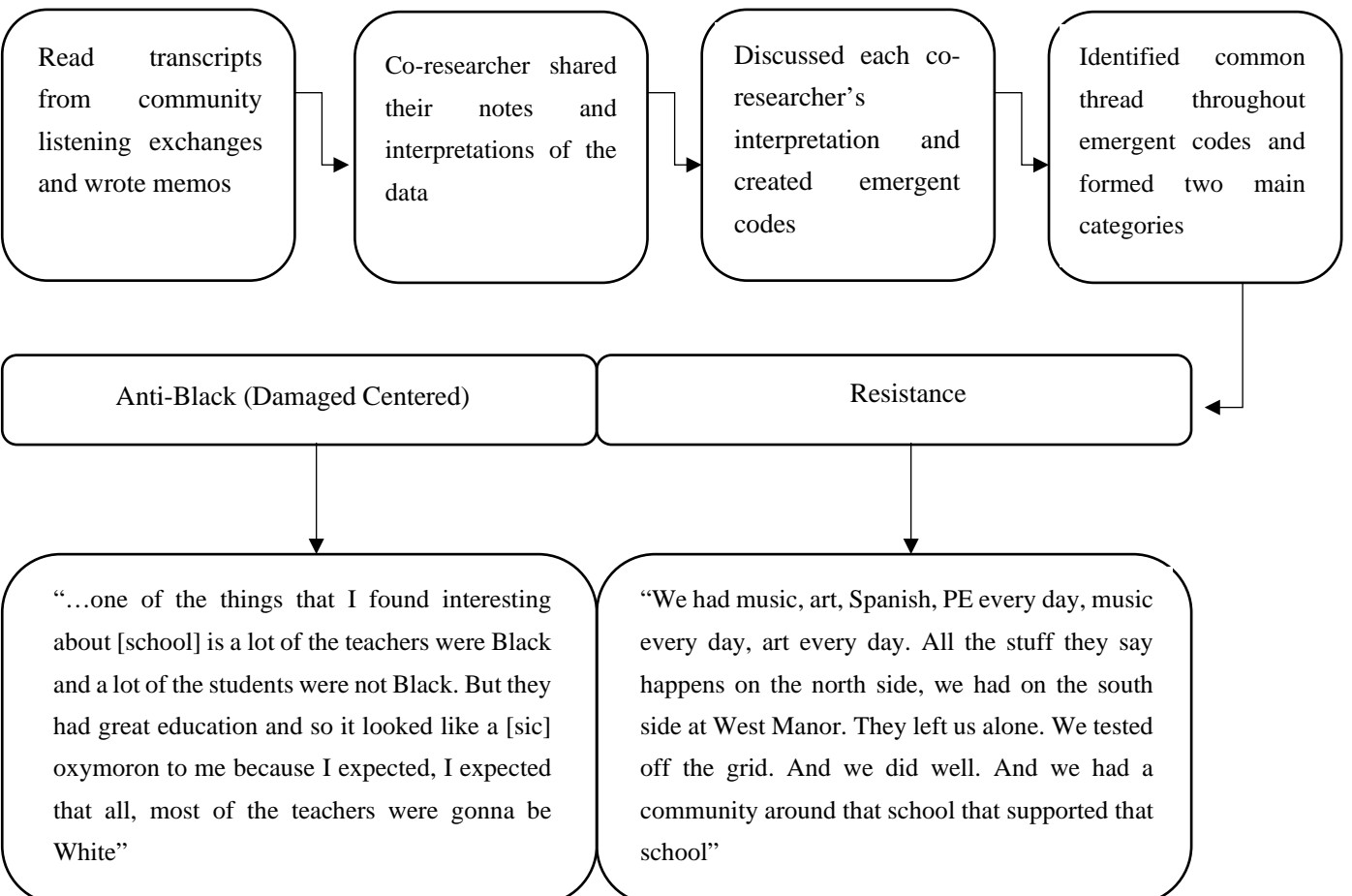

**Figure 5.** Data analysis process.

*3.1. Results: Resistance to Damage-Centered Research*

Our research collective embarked on an inquiry journey to understand the education concerns, housing motivations, and desires of Black people and new residents more intimately. Community Listening Exchanges provided a space for us to listen, learn, and share knowledge with participants. We analyzed transcripts after each listening exchange using an inductive and thematic analysis method, looking for patterns across participant statements and expressed ideals. Two distinct yet interconnected themes emerged from our analysis of the transcripts of the three Community Listening Exchanges: anti-Black discourse and resistance.

3.1.1. Anti-Black Discourse

During the first listening exchange, a Black, male participant remarked, "One of the things that I found interesting about [affluent white school] is a lot of the teachers were Black and a lot of the students were not Black. But they had great education and so it looked like a [sic] oxymoron to me because I expected, I expected that all (pause) most of the

teachers were gonna be White and that wasn't the case." We missed the comment during the listening exchange, but the statement bounced off the page during our data analysis. The participant's comment demonstrates the ways in which deficit-based, dominant narratives label the lives and expectations of Black teachers. The statement of this Black participant did not offer an upbeat assessment of the intellectual and rigorous work Black teachers were conducting in a White school but a total shock and surprise that Black teachers were capable of such academic excellence. This statement is aligned with a response from the second listening exchange. When we asked participants to share where they see themselves within the changes happening in the neighborhood and school, a second listening session participant responded this way: "I think when I first moved into the neighborhood, I had a much greater hope for the changes, and I was very active in the neighborhood. I tried to get to know neighbors. And I had a lot of passion for seeing restoration and redemption in the neighborhood. And although I would never have said it at the time, I'm sure there was a part of me that was being fueled by the great white hope, you know, and trying to save the children . . . my priorities shifted and now I find myself living in the neighborhood, not being involved in it and I don't have a lot of passion like I used to. And so, I almost feel numb." The White female participant's views of the neighborhood and school were also deficit-based in that she wanted to "save the children" as the "great white hope." In her estimation, Black schools and Black neighborhoods were lacking, and there was nothing to be gained from Black schools and neighborhoods because they needed "redeeming" and "restoring." Both participants pitch anti-Black discourse through a dehumanizing deficit lens. Both participants, one Black and one White, saw no value in Black schools or with Black teachers. Their discourse indicates anti-Black rhetoric that openly and casually floats in conversations about education and housing in gentrifying Black neighborhoods.

3.1.2. Resistance

Despite the anti-Black discourse, CLEs provided a space that allowed for stories of resistance and triumph to emerge. During the first listening exchange, a Black woman participant raved about a Southwest Atlanta neighborhood where she grew up and the school she attended: *I grew up in my ideal neighborhood. I grew up where my parents were one of the first Black families to move into Cascade. My parents took over West Manor if you look at the pictures from 1964, West Manor class of 1964 it was an all-white school. By 1972, it was [a] predominantly Black school with Black teachers. APS [Atlanta Public Schools] left us alone. APS never came over to West Manor Elementary School: "Leave those Black kids alone. Because those parents do not play, leave them alone, whatever they want." We had music, art, Spanish, PE every day, music every day, art every day. All the stuff they say happens on the north side, we had on the south side at West Manor. They left us alone. We tested off the grid. And we did well. And we had a community around that school that supported that school.* Her statement and the Pittsburgh history presentation by a local Atlanta historian provided a picture of a Southwest Atlanta community rich in community pride, Black excellence, and deep kinships. Our form of resistance was providing all three audiences with asset-based knowledge of Black neighborhoods and schools. The legacy of Black teachers and Black excellence guided our analysis of such pronouncements in helping to correct the false narrative about Southwest Atlanta neighborhoods and schools.

**4. Discussion**

Education and housing are the two mechanisms that prop up the notion of the American Dream. This notion is out of reach for many in urban cities and neighborhoods. Our journey—our initial research design mistakes and our CLE facilitation—provides a clear example of how researchers can disrupt damage while resisting deficit narratives in community research. Our work as educators collecting, analyzing, and presenting data about the experiences of Black students and communities at the intersection of gentrification and urban education reform is a justice-centered innovation primarily because we are explicit in our goal to conduct research alongside parents and communities to confront

gentrification as a new form of school pushout. Despite initial findings that appeared to highlight anti-Black discourse about participants' perspectives of and experiences in the communities we serve, we used a collective, critical approach to generating and analyzing data that allowed us to uncover the ways in which housing and educational displacement relied on deficit narratives to justify the removal of marginalized people. Our research drew on Gordon da Cruz's (2017) argument for critical community-engaged scholarship. Our intention of facilitating a two-way exchange of give and take was realized by inviting a local historian, sharing articles and book titles, and showing up as teachers and neighbors. We continuously reflected and shifted throughout our process to ensure our aims were justice-centered. We worked collaboratively as co-researchers with parents and community members to illustrate critically conscious knowledge about how their experiences with school and housing were shaped by race and racism. We understood that we were not the experts and relied heavily on the expertise of all participants. Additionally, we were direct in our aim to conduct research for and with Black people. Finally, we worked diligently to highlight the community's assets and share the stories of love, joy, and community that are often left out of research on marginalized groups.

Teachers as researchers for social transformation are largely absent from the research field (Stapleton 2019). First, teachers are often too overworked and overwhelmed with administrative duties to engage in and conduct full-scale research around social issues. Additionally, employment-at-will policies deter teachers from undertaking justice-centered work beyond the scope of their classroom and school duties. Lastly, Black teachers in Atlanta have been publicly criminalized and marginalized for reframing damage-centered narratives pushed by large urban districts (Robinson and Simonton 2019).

There were several limitations and challenges in conducting CLEs. First, because teachers conducting research for social transformation through a critical community-engaged research approach are absent from research literature, we essentially created an innovative justice-centered research method with no guides. Furthermore, conducting CLEs requires an intimate level of community ties and connections for open and authentic conversations. Our TPAR collective successfully conducted CLEs because of our strong community ties and commitment to our neighborhood and schools. For example, Shakale recruited her school's parents, Rebecca invited her coworkers, Shae tapped into her relationship with community newcomers, and Thais leaned on her connections with a community organization to forge spaces for courageous exchanges. This was a limitation because recruiting participants to a "living room" conversation—with no connection to the neighborhood or schools—would have garnered a limited group and limited data. Lastly, time was our most significant limitation for trust building and exchanging. For instance, a participant at the first listening exchange did not say a word until the end of our exchange time. It was as if he was just getting warmed up. During each CLE, participants had more questions, more thoughts to share, more connections to make, and networks to build.

Each CLE also presented a set of challenges. During the CLE, there were issues of power at play. Two community elders were in attendance, and they often clashed in their views. As teacher-researchers, we had to toe the line of respect for community elders and maintain the CLE focus. Another challenge was the very definition of community, as each community we met embodied a different meaning for the word. For example, participants of the second CLE often referred to community, but this often excluded longstanding residents, and participants of the third CLE referenced community focused on their immediate family. To counter this in future CLEs, we think it would be best to conduct norm setting at the outset to center participants on a shared view of community. We also recommend an examination of this dichotomy in future research studies.

Regardless of these limitations and challenges, Community Listening Exchanges offer a multitude of strengths. First, CLEs allowed us to co-produce knowledge with community members, meaning we expected to learn from them to enhance our knowledge and perfect our craft as teachers. Next, CLEs also served as an invitation to Black parents as knowledge co-producers to understand that they are our clients and we serve them and their children.

Finally, Community Listening Exchanges allowed us to foster trust, compassion, and more profound knowledge, resulting in high-quality and authentic data collection and results. Our presence as [Black] teacher-researchers was a disruption and was a radical act especially given the criminalization and politicization of Black teachers during the sensational Atlanta Cheating Scandal. By positioning ourselves as teacher-researchers in living rooms across Southwest Atlanta, we stood on the intellectual and activist legacies of Black teachers before us to highlight sociopolitical and academic knowledge that is often omitted in damage-centered narratives.

**Author Contributions:** Conceptualization, T.C., S.E., S.G. and R.G.; methodology, T.C., S.E., S.G. and R.G.; data analysis, T.C., S.E., S.G. and R.G.; writing—original draft preparation, T.C. and S.E.; writing—review and editing, T.C. and S.E. All authors have read and agreed to the published version of the manuscript.

**Funding:** This research received no external funding.

**Institutional Review Board Statement:** The study was conducted in accordance with the Declaration of Helsinki, and approved by the Institutional Review Board of Georgia State University (IRB number H18562 approved 22 May 2018).

**Informed Consent Statement:** Informed consent was obtained from all subjects involved in the study.

**Data Availability Statement:** Not applicable.

**Acknowledgments:** The authors would like to thank Joyce Elaine King, Benjamin E. Mays Endowed Chair of Urban Teaching, Learning, and Leadership and Professor of Educational Policy Studies in the College of Education and Human Development at Georgia State University, for her tireless support of our study and her unwavering commitment to Black communities and families across the diaspora.

**Conflicts of Interest:** The authors declare no conflict of interest.

## Notes

[1] "Gentrification is urban revitalization driven by profit that results in the displacement of historically marginalized working-class communities and communities of color. Typically, these communities have struggled with too few jobs, amenities, and services because of years of disinvestment. Gentrification is led by private developers, landlords, and businesses, and often happens in areas where land is inexpensive and the potential to turn a profit is high. While development is usually framed as coming from the actions of private businesses, government policy plays a key role in promoting gentrification by offering tax incentives, zoning, and infrastructure improvements. As neighborhoods are developed and renovated, newer housing stock attracts higher-income residents as land value, rents, and property taxes all rise. This in turn can lead to widespread displacement of community members, often low-income people of color, who are priced out. Ironically, while development brings much needed amenities such as schools, commercial districts, and grocery stores, the low-income populations most in need of such services do not reap the eventual rewards of investment" (Housing Justice League and Research Action Cooperative 2017, p. 19).

[2] We are four women, three Black and one Jewish-American, who are committed to being valuable members of the Black communities we serve. Through our commitment, we embody the skills, knowledge, mindsets, consciousness, worldview, beliefs, attitudes, and dispositions, that is to say, souls, required for highlighting liberating and humanizing views of Black students, families, and communities (Council 2021). We argue that all teachers who are committed to working with Black students and communities possess these skills to be a teacher in an era where educational, social, political, cultural, environmental, and economic challenges are multiple and layered. Additionally, the premise of enclosing Black in brackets is to signal to the reader, if we remove the word Black, teachers should ideally possess the knowledge, skills, mindset, consciousness, worldview, beliefs, attitudes, and dispositions, or souls to conduct critical community-engaged work in Black and urban communities. To be a [Black] teacher then requires a humanizing, asset-based, collaborative disposition to actualize liberty and justice for students, families, and communities.

[3] Georgia recognizes the doctrine of employment at will. Employment at will means that in the absence of a written contract of employment for a defined duration, an employer may terminate an employee for good cause, bad cause, or no cause at all, so long as it is not an illegal cause (Georgia Secretary of State, sos.ga.gov/index.php/corporations/what_georgia-employers_need_to_know, accessed on 4 November 2020).

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
