# Peer review of "[Black] Teachers Resisting Damaged-Centered Research: Community Listening Exchanges as a Reciprocal Research Tool in a Gentrifying City"

_socsci, doi:10.3390/socsci11020084_

Round 1
Reviewer 1 Report
Thank you for the opportunity to read and evaluate this interesting and well-written manuscript.
The paper, which details the efforts of a group of educators to carry out the development of community listening exchanges as a PAR method in a gentrifying southwest Atlanta, offers a critical counterpoint to the ongoing oppression happening in educational circles in the US. As I write, a number of state governments are placing severe constraints on the discussion of race, racism, and inequality in many classrooms—in effect, asking educators to ignore the social and spatial realities of their students and communities. I wonder if it might be helpful to make explicit reference to the CLE method in relation to the ongoing this educational crisis regarding race.
This submission demonstrates in an effective way how listening and responding to the needs and issues of communities is a pedagogical responsibility along with pushing back again the deficit- and damage-based research that has come to define academic and public thinking about Black communities in the US. I appreciated the emphasis the teachers placed on developing the spaces and practices necessary for conducting an asset-based approach to the community listening exchanges.
I liked the transparency and reflexivity that runs throughout the paper. The submission allows the educators to speak as a collective while also allowing us to hear the voices of individual authors and of course specific community member participants. Especially helpful from a methodological perspective was the way the authors discussed the failure and unsatisfactory ethics and power dynamics of their original plan of individual semi-structured interviews and how that realization led to the community listening exchanges. The distinction the authors make between CLEs and traditional focus groups was helpful as well. All of this along with many background statements in the paper sets the stage for readers to replicate and apply community listening exchanges in their own schools and communities.
I am strongly supportive of the paper’s publication, although I would ask that the authors address three issues—one of which is major and the other two are minor but important. In terms of minor points, would the authors tell us when (the date) of Techwood’s construction as they discuss the “twice-cleared community” idea? Also, in the presentation reproduced from Nasir Muhammad, the authors suggest that Sherman burned Atlanta in 1917. I am confused by this, Sherman burned Atlanta during the Civil War—correct? In terms of major revision, I would like to see the authors discuss or list some of the challenges and possible pitfalls facing community listening exchanges, even as we recognize the value of this methodology. I also wonder how the insight generated in community listening exchanges found their way into and informed educational practices in the schools or communities. The authors stress how they sought and achieved social transformation, but we don’t hear as much in that area. In other words, how was the listening put to work and what did it generate?
Author Response
February 4, 2022
Social Sciences Journal
MDPI OPEN ACCESS PUBLISHING ROMANIA SRLStr Avram Iancu 454
Floresti, Cluj, Romania
Dear Colleagues,
Thank you for the opportunity to publish our manuscript, “Community Listening Exchanges: [Black] Teachers Disrupting Damage-Centered Research where Gentrification is the New School Pushout” in the New Trends in Community-Engaged Research: Co-producing Knowledge for Justice special issue of the Social Sciences Journal. We are grateful for the reviewers’ thorough and careful feedback.
Reviewer #1
Thank you for your glowing compliments of our manuscript. Please see our responses to your feedback in the non-bolded text below:
Make explicit reference to CLE method in relation to the ongoing educational crisis regarding race.
We have strengthened the introduction to make race more explicit from the outset. By doing this, the reader can see how race runs through the entire manuscript and how race is the central component of our intentional and innovative research method.
Would the authors tell us when (the date) of Techwood’s construction as they discuss the “twice-cleared community” idea?
Thank you for highlighting this omission for us. We have included the year construction started on Techwood Homes, 1934, on page 3, in section 2.4, paragraph 2.
Also, in the presentation reproduced from Nasir Muhammad, the authors suggest that Sherman burned Atlanta in 1917. I am confused by this, Sherman burned Atlanta during the Civil War—correct?
Thank you for highlighting this misprint. Nasir Muhammad, historian and owner-operator of Black Mecca of the South Tours, reviewed his presentation transcript and corrected this slight error to reflect the following: Now, another thing that Pittsburgh does not get credit for is the refuge the community provided after the Atlanta Massacre in 1906. Don’t let anyone ever tell you that Sherman burning down Atlanta during the Civil War was the worst thing that ever happened in terms of fire. No! That’s not the worst fire in Atlanta’s history. The Great Atlanta Fire of 1917 from Edgewood Avenue to Ponce de Leon Avenue left 10,000 Black people homeless. The Pittsburgh community opened their doors. Pittsburgh churches, communities, and organizations allowed Black families from the Old Fourth Ward to come and live.
In terms of major revision, I would like to see the authors discuss or list some of the challenges and possible pitfalls facing community listening exchanges, even as we recognize the value of this methodology.
We responded to this feedback by adding the limitations and challenges of conducting CLEs to the discussion section of the manuscript.
I also wonder how the insight generated in community listening exchanges found their way into and informed educational practices in the schools or communities. The authors stress how they sought and achieved social transformation, but we don’t hear as much in that area. In other words, how was the listening put to work and what did it generate?
In response to this request, we added the section 2.5 titled, “Resisting Damage-Centered Research Beyond Community Listening Exchanges” to offer readers insight into the additional actions/activism of our research.
Thank you again for your time and service. We are immensely grateful!
Respectfully,
Manuscript Co-Authors

Reviewer 2 Report
This “research and activism” is critical and a much-needed valuable contribution. It was a pleasure to read and learn from this work! The offering of CLEs as a reciprocal research tool is a contribution and the focus on the work and legacy of black teachers is invaluably important. While I appreciate this article, I recommend some significant revisions to make it stronger for publication.
Overall, my major feedback to consider is that this reads as 2-3 articles at least. All 2-3 of those articles are valuable and should be published eventually, but this piece needs more clear focus throughout. The title and organization of the article can, and should, be used to clearly lead the reader through to the most critical arguments. There is a reference to a larger study and it is unclear how this article fits with that and other publications that have come from the study. Perhaps more clear explanation of that and citations to other work the authors have published could help the researchers remove some peripheral topics from this article and focus.
Ultimately my question is which of these topics mentioned in the article is this really about: Is this a methods article/contribution? Is this an autoethnographic investigation of choosing a research method? Is this about gentrification? Is this about education and housing more broadly? Is it about resisting deficit narratives? Is it about anti-blackness and resistance? Is it about the motives and desires or black people? Of newcomers? Is it about black teachers and their roles? Is it about gentrification as the new school push-out? Is it about confronting gentrification? Is it about sharing stories of love? Is it about the school system, real estate holders, and developers collaborating to create a neighborhood free of poor, Black people? If it is about all of these, then they all need a lot more flushing out and connection ...and if not, I would suggest separating this work into different articles and working to clarify the organization and contribution of this particular article. My comments here would vary depending on what the researchers decide to make this particular article about. Whichever focus is chosen, it needs more literature review, more methods, a discussion of limitations, and more findings and analysis for that particular topic of focus.
What is the large finding/overarching takeaway of this work? To me a very powerful part of this article is the discussion about the original research methods and the change the researchers made. This needed more space and more organization and more about the eventual CLE methods and analysis and limitations. Or, if the focus of this article is really more about school pushout and gentrification, perhaps the methods process needs less space and more detail in an appendix (not because it is not valuable, but because there is not enough space here to do all the authors claim to do here). On page 15- it says of this work, “allowed us to uncover the ways in which housing and educational displacement relied on deficit narratives to justify the removal of marginalized people.” This paper does not go deep into uncovering the connection between these narratives and displacement, there are a few quotes and some data about the changing neighborhood, but far more needs to be done/presented to uncover this for the reader. On page 13 it says “We also developed a deeper, more critical lens of our teaching, learning, and leadership practices”-how? How is this connected? Explain. On page 16 it says this work “share the stories of love, joy and community that are often left out of research on marginalized groups” but where were these stories and voices of love and joy and community in this article? There was a strong sense of love and community coming from the teacher researchers but not enough from the community if that is the focus. This sentence: “We too could not stand idly by witnessing the displacement, erasure, and removal of our neighborhood and school” is excellent and could help pull the work together throughout, but the researchers need to make clear how their work actually connects to/teaches us new information about displacement of the neighborhood and school.
The authors are clearly critical of other research and qualitative methods and do an excellent job discussing their original study and how and why they changed it. However, they present no limitations or critique to their final method and analysis. They are clearly reflexive practioners and should include some reflection on the final methods used. On page 2 they say this will “eliminate the hegemonic gaze.” Is elimination in this work ever truly possible even with incredibly thoughtful qualitative methodologies like CLEs? Perhaps “working to eliminate” would be more accurate? I appreciate on page 2-3 the authors discuss telling the story from the “ground up” and sharing “similar experiences,” but is this truly and fully ground up? Are the experiences all similar for all? The authors could spend more time considering power, title, cultural and social capital in the CLEs and analysis. The authors discuss critiquing literature and the gaps in the literature, but include little to no citations of other qualitative methods/PAR in urban education and in youth and gentrification work (of which there is a lot).
Smaller/more specific comments:
-Love the inclusion of “collective love” as a propelling force-wonderful!
-Section 2.1 could use more demographic information, it jumps around. What are the racial demographics? Race and SES? What are the schools like demographically?
-Footnote 1 is helpful in explaining the use of [Black] but should be clarified more and the explanation of the larger research project here seems confusing and should be in a methods section instead and explained more
-Page 3 says this was “an inquiry journey to understand the motivations and desires of Black people and new residents.” This is very broad-does this article and the research presented in it do that? Does other work from this study? Later it says, “Our inquiry process started by using our firsthand school and classroom experiences to list the issues plaguing Black students and schools.” This seems like a clearer starting place for the work featured in this article.
-Needs more background/data on gentrification in the neighborhood? Demographics of school community to show the pushing out?
-“All of the markers of gentrification surrounded us” citations? What are the markers? How do the authors define gentrification? What is their stance on gentrification? What literature do the researchers rely on to inform their views of gentrification? How did they present it in each CLE?
-“We started by reviewing literature on the topic. We found that the current literature did not offer a glimpse into the intersections of urban education reform and gentrification through a teacher inquiry.” While there may not be a study just like this one, there are a lot of studies on related topics and with related methods not referenced here that should be.
-Each researcher is identified by a name and story but don’t mention their race in that section
-“There were institutional agreements to which we had to adhere” -what institution? What was each person’s role and connection?
-“Our initial design was a mistake. Our semi-structured interview approach amounted to unintentionally presenting ourselves as social workers instead of thoughtful listeners and intentional researchers.” This is an incredibly powerful statement and admission. However, it was unclear if the researchers did that protocol or just planned to? Did they actually see first hand the issues with it or just come to understand its flaws on paper? If so, did they collect and use any of that data? Or is this story just included to show us why CLE was used instead? Just be clear about how this connected to the study presented here.
-Page 5 “We expected participant responses to provide different contexts for the effects of urban education reform and gentrification.”-unclear what this wording means here.
- The discussion of reflexivity is a powerful contribution!
-Throughout there is a lot of mention of literature but very little cited and no reference to other work that has similarly privileged knowledge of community
-The CLE method part needs a lot more information. Who participated in CLEs? Who didn't? Can you provide race/class/age/gender breakdowns in a chart? How did your role as teachers factor in? Were these people connected to the schools? How was recruitment done for all of them? Why the decision to focus a whole CLE on a charter school community? There's no other mention of role of charters etc... On page 16 it says, “We also invited parents as knowledge co-producers with the understanding that they are our clients and we serve them and their children.” Were these families of children the authors serve in the CLEs? The article also says of the researchers “so that their identity and privacy would not be compromised”-this feels confusing - what is meant by this? What did they share, what did others share, how did/did privacy conflict with “neighbors first” and the role of “serve them and their children”?
-On page 8 “in-town neighborhood of the research site in the living and dining room of a renovated house listed for sale” confusing wording here
-Situate this in time throughout
-Again more is needed on the methods and the cosponsors? What was the relationship with a broker? What were participants and sponsors told about the study? How did the researchers choose articles and materials to present for each audience? Why did they change for each? Why was no presentation and information shared at the 3rd one? How then is this different from a more traditional focus group? How does the role of researcher as community member change with each different group?
-“Giving more than we took”-appreciate this, excellent work
Author Response
February 4, 2022
Social Sciences Journal
MDPI OPEN ACCESS PUBLISHING ROMANIA SRLStr Avram Iancu 454
Floresti, Cluj, Romania
Dear Colleagues,
Thank you for the opportunity to publish our manuscript, “Community Listening Exchanges: [Black] Teachers Disrupting Damage-Centered Research where Gentrification is the New School Pushout” in the New Trends in Community-Engaged Research: Co-producing Knowledge for Justice special issue of the Social Sciences Journal. We are grateful for the reviewers’ thorough and careful feedback.
Reviewer #2
Thank you for your thorough review of our manuscript. Your critical eye really helped us to see several gaps in our writing. We have given your feedback much thought and incorporated many of your suggestions and highlights. Our responses are below in non-bolded text:
Footnote 1 is helpful in explaining the use of [Black] but should be clarified more and the explanation of the larger research project here seems confusing and should be in a methods section instead and explained more
Thank you for highlighting footnote 1. We have updated this footnote to clearly indicate for the reader why we bracket Black in our manuscript. Additionally, instead of using our word limit to go into greater detail of the original study, we simply cite it as source.
Needs more background/data on gentrification in the neighborhood? Demographics of school community to show the pushing out?
We completely agree with your feedback. However, we paid considerable attention to the number of questions you posed regarding the focus of the article. Because this is a methods article, detailing an innovative qualitative research method for critical community-engaged research, we spoke less about gentrification as “pushout” and more about how the research method made residents feel included in a gentrifying climate.
“All of the markers of gentrification surrounded us” citations? What are the markers? How do the authors define gentrification? What is their stance on gentrification? What literature do the researchers rely on to inform their views of gentrification? How did they present it in each CLE?
“We started by reviewing literature on the topic. We found that the current literature did not offer a glimpse into the intersections of urban education reform and gentrification through a teacher inquiry.” While there may not be a study just like this one, there are a lot of studies on related topics and with related methods not referenced here that should be.
Throughout there is a lot of mention of literature but very little cited and no reference to other work that has similarly privileged knowledge of community.
We reviewed the number of times we referenced “literature” in the manuscript and your feedback highlighted for us that we did not in fact cite the literature we reviewed. We have included select citations of the literature on the intersection of urban education reform and gentrification (the focus of our teacher inquiry) we reviewed, particularly in the sentences you’ve highlighted. We would also like to invite readers to reference, section 2.1 “Atlanta as our site for research,” where we briefly highlight data surrounding inequality in Atlanta. We argue that these inequalities have impacted the Southwest Atlanta community in ways that impact homeownership, hence, inviting gentrification.
“Our initial design was a mistake. Our semi-structured interview approach amounted to unintentionally presenting ourselves as social workers instead of thoughtful listeners and intentional researchers.” This is an incredibly powerful statement and admission. However, it was unclear if the researchers did that protocol or just planned to? Did they actually see first hand the issues with it or just come to understand its flaws on paper? If so, did they collect and use any of that data? Or is this story just included to show us why CLE was used instead? Just be clear about how this connected to the study presented here.
Thank you for your questions. We included this narrative to show the readers how easy it is to make unintentional mistakes during critical community-engaged research. We also wanted to explicate for the reader a level of reflexivity one should have when conducting community research. In our estimation, by explicitly stating, “We often consulted elders and our community partner, Gwinnett SToPP, for guidance and feedback,” shows the reader just how intentional we are about our community work. Additionally, it is also commonplace in Black communities for juniors to confer with trusted elders when they are embarking on new journeys or are unsure. This is an effort to make sure we get it right from the start. This step in the process is one that should not be overlooked.
Page 5 “We expected participant responses to provide different contexts for the effects of urban education reform and gentrification.”-unclear what this wording means here.
Thank you! We rewrote the sentence entirely.
The CLE method part needs a lot more information. Who participated in CLEs? Who didn't? Can you provide race/class/age/gender breakdowns in a chart? How did your role as teachers factor in? Were these people connected to the schools? How was recruitment done for all of them? Why the decision to focus a whole CLE on a charter school community? There's no other mention of role of charters etc...
Again more is needed on the methods and the cosponsors? What was the relationship with a broker? What were participants and sponsors told about the study? How did the researchers choose articles and materials to present for each audience? Why did they change for each? Why was no presentation and information shared at the 3rd one? How then is this different from a more traditional focus group? How does the role of researcher as community member change with each different group?
On page 16 it says, “We also invited parents as knowledge co-producers with the understanding that they are our clients and we serve them and their children.” Were these families of children the authors serve in the CLEs? The article also says of the researchers “so that their identity and privacy would not be compromised”-this feels confusing - what is meant by this? What did they share, what did others share, how did/did privacy conflict with “neighbors first” and the role of “serve them and their children”?
Thank you! We have revised the sections labeled, Community Listening Exchange 1, Community Listening Exchange 2 and Community Listening Exchange 3 to include the points you raised in your questions. We also included participant race/age range/gender breakdowns in Tables 1, 2, and 3. Also, we believe the entire paper details our roles as teachers in conducting Community Listening Exchanges. Additionally, we inserted information regarding recruitment for each CLE. Lastly, we added more information to explain CLEs’ connections to neighborhoods and schools throughout the manuscript.
On page 8 “in-town neighborhood of the research site in the living and dining room of a renovated house listed for sale” confusing wording here
Thank you! We have updated this wording to read more clearly.
“There were institutional agreements to which we had to adhere” -what institution? What was each person’s role and connection?
We were simply detailing that we completed an Institutional Review Board process as teacher researchers. We’ve updated the language to clearly articulate this.
Page 3 says this was “an inquiry journey to understand the motivations and desires of Black people and new residents.” This is very broad-does this article and the research presented in it do that? Does other work from this study? Later it says, “Our inquiry process started by using our firsthand school and classroom experiences to list the issues plaguing Black students and schools.” This seems like a clearer starting place for the work featured in this article.
Essentially, these two sentences are the same as we state in each sentence that our starting point was the same with each mention of an inquiry project: “to understand the education concerns, housing motivations, and desires of Black people and new residents more intimately.”
Section 2.1 could use more demographic information, it jumps around. What are the racial demographics? Race and SES? What are the schools like demographically?
Thank you for highlighting this as “more background/data in the neighborhood” is one of the areas we initially included in the paper but we have strongly considered your many questions about the focus of this paper and collectively, we landed on focusing on the CLEs’ process as a qualitative research method. Please see para 1 in section 2.1 as this is where we briefly detail much of the background on gentrification in SW Atlanta without making this a 2-3 article manuscript.
Each researcher is identified by a name and story but don’t mention their race in that section.
Because our races are mentioned previously in the introduction and because Rebecca is the only co-researcher that experienced “white flight,” we believe the reader can deduce that she is the Jewish American woman and the remaining co-researchers are Black women.
Thank you again for your time and service. We are immensely grateful!
Respectfully,
Manuscript Co-Authors

Round 2
Reviewer 1 Report
Thank you for the opportunity to read and evaluate this revised version of the manuscript. I believe the authors have responded to my earlier suggestions and criticisms in a thorough and professional way. I am satisfied that the paper is now ready for publication.
Author Response
Dear Colleagues,
Thank you for the opportunity to publish our manuscript, “[Black] Teachers Resisting Damaged-Centered Research: Community Listening Exchanges as a Reciprocal Research Tool in a Gentrifying City” in the New Trends in Community-Engaged Research: Co-producing Knowledge for Justice special issue of the Social Sciences Journal. We are grateful for the reviewers’ thorough and careful feedback.
Reviewer #1
Thank you for your glowing compliments of our manuscript. We are delighted and grateful that our revisions passed your review.
Reviewer 2 Report
These improvements were excellent and it was exciting to see such progress. I look forward to this article being published. I have additional thoughts for improvements and clarifications, however.
-The title change signals major improvement to the focus of the paper, well done!
-This section feels out of place/under explained: “Our neighborhood is riddled with poachers – those who lurk from the outside, wait for the prime opportunity (read: return on investment potential), and then pounce on minoritized and marginalized residents who have begged city officials for resources for decades. The poachers set up shop, dehumanize and criminalize the survival tactics of longstanding residents and ultimately participate in an economic system to push them out.” Are there citations or evidence/examples to support this addition? It feels out of place and should be integrated into the flow of that section more clearly. This sentence later on feels very clear: “All of the markers of gentrification surrounded us: changing demographics of our schools, letters from landlords and investors, and postcards from newcomers who saw Black children as a threat”- the one in the introduction could be more similar to this to set the background and the researchers role in it.
-The authors still need to define gentrification somewhere early on (maybe even footnote the definition you gave participants), not everyone defines this the same way.
-The literature cited on urban education reform and gentrification feels incomplete with many older references. While not perfect matches see Quarles & Butler, 2018 lit review for some more recent work: https://www.tandfonline.com/doi/abs/10.1080/0161956X.2018.1488399
Also can see Butler, 2021; Cordova-Cobo, 2019; Makris & Brown, 2017; Stillman, 2012; Makris, 2015; Freidus, 2019; Cucchiara, 2013; Posey-Maddox, 2014; Pearman, 2019 etc... to dive into some of this work.
-Good clarifications and changes around the decision to do CLE not interviews, but the authors should still clarify that they did not actually do the interviews (right?) that they just had the protocol then switched methods after discussion with elders.
-How did you manage to recruit different demographic audiences for 1 and 2, more specifics needed.
-I still want a lot more clarity throughout around the researchers role as teachers during, and in, this study. It sounds at times like the researchers did not use this connection for recruitment or reveal this all to participants, as it says “All sessions were held at a neutral location away from teachers’ work environments so that their identity and privacy would not be compromised.” But then it says “We encounter these parents often through our school interactions at Gideons Elementary School and their voices are often dismissed.” And then later “and showing up as teachers and neighbors”-how did the researchers show up as teachers? Explain. Then again later it says the researchers did recruit from schools. The authors say “teacher researchers”, but how are they acting as teacher researchers vs teachers who are also researchers (which there are many in the lit and many who do PAR)? This could just be spelled out more.
-“Most of them had enrolled their children in charter schools or other schools that they created amongst their networks”-this leaves me wanting a lot more information and also curious how the teacher researchers grapple with the role of charter schools and gentrification! I know there is not space for this here but there is no mention of this or the lit on gentrification and school choice/charters.
-I think the new focus is very strong and appreciate the part showing the impact of this work on groups of people, it says “We share this to illustrate how the insight generated in CLEs found their way into and informed educational practices in schools and communities in Atlanta.” I do think we could use one or two sentences prior to this section that summarize what that overall insight was very briefly. What was the big take away from these 3 CLEs? The data analyses section after this is confusing and hops around to this. In general the organizational flow could be clarified some, particularly in sections 2 and 3. I think the parts of section 3 with the findings of the CLE could go before the authors talk about the impact of their work. I think the take aways/findings about anti-black discourse and resistance should be clarified- briefly tell the reader the big takeaway and why it is new and important information from your CLE that could then inform these practioners. How does this inform policy/action/activism?
-The limitations don’t discuss the ways you may have a particular sample given your recruitment methods . “Furthermore, our TPAR collective was successful in conducting CLEs because of our strong community ties and commitment to our neighborhood and schools,” but how is this also a limitation? Explain.
-“Another challenge was the very definition of community as each community we met embodied a different meaning for the word. For example, participants of the second CLE often referred to community, but this often-excluded longstanding residents and participants of the third CLE referenced community focused on their immediate family. To counter this in future CLEs, we think it would be best to conduct norm-setting at the outset to center participants on a shared view of community.”-this seems like a very interesting finding in and of itself for another paper perhaps!
-Excellent last line!
Author Response
Reviewer #2
Thank you for your thorough second review of our manuscript. Your critical eye really helped us to see additional gaps in our writing. We have given your feedback much thought and incorporated many of your suggestions and highlights. Our responses are below in non-bolded text:
|
Review Feedback/Comments |
Co-Authors explanation |
|
These improvements were excellent and it was exciting to see such progress. I look forward to this article being published. I have additional thoughts for improvements and clarifications, however. |
Thank you! |
|
-The title change signals major improvement to the focus of the paper, well done! |
Thank you! |
|
-This section feels out of place/under explained: “Our neighborhood is riddled with poachers – those who lurk from the outside, wait for the prime opportunity (read: return on investment potential), and then pounce on minoritized and marginalized residents who have begged city officials for resources for decades. The poachers set up shop, dehumanize and criminalize the survival tactics of longstanding residents and ultimately participate in an economic system to push them out.” Are there citations or evidence/examples to support this addition? It feels out of place and should be integrated into the flow of that section more clearly. |
We revised these sentences to boldly depict our experiences with gentrification. The revision currently reads: “This paper details Community Listening Exchanges (CLEs), in which we examined the intersection of urban education reform and gentrification in Southwest Atlanta. Our Southwest Atlanta neighborhood and schools are riddled with poachers – those who lurk from the outside, wait for the prime opportunity (read: return on investment potential), and then pounce on minoritized and marginalized residents who have begged city officials for resources for decades. This is how some in our neighborhood and schools experience gentrification. As community-centered teachers, we too witness and experience gentrification this way. Through our collaborative research, we decided to stand in the gap of urban education reform and gentrification even as it impacts our personal and professional lives. It is with this lived experience that we approach research in our communities.”
Additionally, evidence is sprinkled throughout the manuscript as we talk about the markers of gentrification in our neighborhood and schools and we seek to enact social transformation for our students while conducting research through the lens of our communities without a hegemonic gaze.
We know that this is strong language and we believe that this is the point of such a strong statement - to draw in the reader - and to show how CLEs disrupt the actual, physical and psychological damage residents endure.
Because this is our actual experience, we did not want to continue in the spirit of neighborhood poachers during CLEs. The manuscript provides details of our process and how intentional we were as teachers and neighbors. |
|
This sentence later on feels very clear: “All of the markers of gentrification surrounded us: changing demographics of our schools, letters from landlords and investors, and postcards from newcomers who saw Black children as a threat”- the one in the introduction could be more similar to this to set the background and the researchers role in it. |
Thank you! We incorporated your feedback into the introduction. Please see the response above. |
|
-The authors still need to define gentrification somewhere early on (maybe even footnote the definition you gave participants), not everyone defines this the same way. |
We responded to your feedback by defining gentrification through a footnote in the introduction section. |
|
-The literature cited on urban education reform and gentrification feels incomplete with many older references. While not perfect matches see Quarles & Butler, 2018 lit review for some more recent work: https://www.tandfonline.com/doi/abs/10.1080/0161956X.2018.1488399 |
Thank you for sharing these sources. We have updated the manuscript with more complete and current references, specifically, Quarles & Butler, 2018; Cucchiara et al, 2014; and Cordova-Cobo, 2019. Also note, in the sentences where we describe our democratic inquiry process, we referenced current literature at the time of study which was 2018. |
|
Also can see Butler, 2021; Cordova-Cobo, 2019; Makris & Brown, 2017; Stillman, 2012; Makris, 2015; Freidus, 2019; Cucchiara, 2013; Posey-Maddox, 2014; Pearman, 2019 etc... to dive into some of this work. |
Thank you again for sharing these contributions to urban education and gentrification. The Quarles & Butler, 2018 article is a great suggestion and one that validates the focus and goals of our research. As such, we have cited this article to show our scholarship is still relevant. |
|
-Good clarifications and changes around the decision to do CLE not interviews, but the authors should still clarify that they did not actually do the interviews (right?) that they just had the protocol then switched methods after discussion with elders. |
We responded to this feedback on page 6, which now reads “We proposed a Community Listening Exchange instead of interviews or surveys because we saw it as a reciprocal data collection method where participants would provide us with data and we, in turn, would share information and resources with them.”
Please note, we also added this sentence at the end of the paragraph on page 5: “Ultimately, we did not conduct interviews and decided to consult our elders instead.” |
|
-How did you manage to recruit different demographic audiences for 1 and 2, more specifics needed. |
We responded to this feedback in the following sections:
Community Listening Exchange 1 - We added 5 words to the end of this sentence: The first exchange took place in the open concept living and dining room of a renovated house listed for sale in Pittsburgh, Southwest Atlanta, a historic, working-class, Black neighborhood.
Community Listening Exchange 2 - The second listening exchange was in the living room of a White family new to Washington Park, Southwest Atlanta. This family had been in the neighborhood since 2009. Lee [pseudonym] worked as a missionary and basketball coach at nearby Booker T. Washington High School – an all-Black, historic neighborhood school where Dr. Martin Luther King Jr. matriculated - and knew one of the co-researchers since her teenage years. Lee also works with a local foundation. Lee recruited her circle of friends and neighbors using her strong neighborhood ties.
I also want to highlight again that after each iteration of data analysis, we discovered missing demographic groups and decided to host additional CLEs to reach data saturation. See this sentence in section page paragraph |
|
-I still want a lot more clarity throughout around the researchers role as teachers during, and in, this study. It sounds at times like the researchers did not use this connection for recruitment or reveal this all to participants, as it says “All sessions were held at a neutral location away from teachers’ work environments so that their identity and privacy would not be compromised.” But then it says “We encounter these parents often through our school interactions at Gideons Elementary School and their voices are often dismissed.” And then later “and showing up as teachers and neighbors”-how did the researchers show up as teachers? Explain. Then again later it says the researchers did recruit from schools. The authors say “teacher researchers”, but how are they acting as teacher researchers vs teachers who are also researchers (which there are many in the lit and many who do PAR)? This could just be spelled out more. |
To provide the reader with a clearer context of our “neutral” locations, we responded to your feedback by replacing this sentence: “All sessions were held at a neutral location away from teachers’ work environments so that their identity and privacy would not be compromised.” with these sentence: In our roles as teacher researchers, we were not immune to the threat of being negatively impacted by Georgia’s employment at-will policies3. To be present as teacher researchers while understanding threats to livelihood, Aall sessions were not held in school buildings but held at a neutral location to allow teachers and participants to speak freely. |
|
-“Most of them had enrolled their children in charter schools or other schools that they created amongst their networks”-this leaves me wanting a lot more information and also curious how the teacher researchers grapple with the role of charter schools and gentrification! I know there is not space for this here but there is no mention of this or the lit on gentrification and school choice/charters. |
We strongly considered your feedback from the first review. We are leaning on our citations, our transcripts detailing our democratic inquiry process, and the information we were able to include given the focus of the manuscript to show how we grappled with the role of chart schools and gentrification. |
|
-I think the new focus is very strong and appreciate the part showing the impact of this work on groups of people, it says “We share this to illustrate how the insight generated in CLEs found their way into and informed educational practices in schools and communities in Atlanta.” I do think we could use one or two sentences prior to this section that summarize what that overall insight was very briefly. What was the big take away from these 3 CLEs? |
Thank you! Instead, we highlighted for the reader how our data analysis was a large part of our study. During data analysis, we discovered the richness and fullness and uninhibited nature of our CLEs. It is with this understanding that we were able to critically examine and highlight the antiblack discourse alongside narratives/acts of resistance and disruption. |
|
The data analyses section after this is confusing and hops around to this. In general the organizational flow could be clarified some, particularly in sections 2 and 3. I think the parts of section 3 with the findings of the CLE could go before the authors talk about the impact of their work. I think the take aways/findings about anti-black discourse and resistance should be clarified- briefly tell the reader the big takeaway and why it is new and important information from your CLE that could then inform these practioners. How does this inform policy/action/activism? |
In the Discussion section, we’ve already accomplished this with the following sentences:
Our work as educators collecting, analyzing, and presenting data about the experiences of Black students and communities at the intersection of gentrification and urban education reform is a justice-centered innovation primarily because we are explicit in our goal to conduct research alongside parents and communities to confront gentrification as a new form of school pushout. Despite initial findings that appeared to highlight deficit information about participants’ perspectives of and experiences in the communities we serve, we used a critical approach to generating and analyzing data that allowed us to uncover the ways in which housing and educational displacement relied on deficit narratives to justify the removal of marginalized people. |
|
-The limitations don’t discuss the ways you may have a particular sample given your recruitment methods . “Furthermore, our TPAR collective was successful in conducting CLEs because of our strong community ties and commitment to our neighborhood and schools,” but how is this also a limitation? Explain. |
We added more details to this paragraph to contextualize for the reader how our strong community ties and commitment to our neighborhood and schools poses as a limitation to those attempting to replicate the study. The revised sentences reads: “Our TPAR collective was successful in conducting CLEs because of our strong community ties and commitment to our neighborhood and schools. For example, Shakale recruited her school’s parents, Rebecca invited her coworkers, Shae tapped into her relationship with community newcomers, and Thais leaned on her connections with a community organization to forge spaces for courageous exchanges. This was a limitation because recruiting participants to a "living room" conversation – with no connection to the neighborhood or schools - would have garnered a limited group and limited data” |
|
-“Another challenge was the very definition of community as each community we met embodied a different meaning for the word. For example, participants of the second CLE often referred to community, but this often-excluded longstanding residents and participants of the third CLE referenced community focused on their immediate family. To counter this in future CLEs, we think it would be best to conduct norm-setting at the outset to center participants on a shared view of community.”-this seems like a very interesting finding in and of itself for another paper perhaps! |
Thank you! We added this line to end of this paragraph: We also recommend an examination of this dichotomy in future research studies. |
|
-Excellent last line! |
Thank you! |
Thank you again for your time and service. We are immensely grateful!
Respectfully,
Manuscript Co-Authors
